# Guided morphogenesis through optogenetic activation of Rho signalling during early *Drosophila* embryogenesis

Emiliano Izquierdo [1], Theresa Quinkler[1] & Stefano De Renzis [1]

During organismal development, cells undergo complex changes in shape whose causal relationship to individual morphogenetic processes remains unclear. The modular nature of such processes suggests that it should be possible to isolate individual modules, determine the minimum set of requirements sufficient to drive tissue remodeling, and re-construct morphogenesis. Here we use optogenetics to reconstitute epithelial folding in embryonic *Drosophila* tissues that otherwise would not undergo invagination. We show that precise spatial and temporal activation of Rho signaling is sufficient to trigger apical constriction and tissue folding. Induced furrows can occur at any position along the dorsal–ventral or anterior–posterior embryo axis in response to the spatial pattern and level of optogenetic activation. Thus, epithelial folding is a direct function of the spatio-temporal organization and strength of Rho signaling that on its own is sufficient to drive tissue internalization independently of any pre-determined condition or differentiation program associated with endogenous invagination processes.

[1] EMBL Heidelberg, Meyerhofstrasse 1, 69117 Heidelberg, Germany. Correspondence and requests for materials should be addressed to S.D.R. (email: stefano.derenzis@embl.de)

Traditional genetic approaches have played a pivotal role in establishing the requirement of individual gene activities and cell behaviors in complex morphogenetic processes[1–9]. More recent advances in synthetic biology are opening the possibility to engineer gene circuits[10], signaling systems[11,12], and biomaterials[13,14] not only to probe morphogenesis, but also to reconstruct it and direct it[15,16]. These approaches, which are converging into the nascent field of synthetic morphogenesis[17], will be instrumental to define the minimum set of requirements sufficient to drive morphogenesis, and will therefore, also facilitate the building of artificial tissues for potential applications in regenerative medicine. Here, we used optogenetics to synthetically reconstitute morphogenesis in the early Drosophila embryo. We focused on epithelial folding, a conserved morphogenetic process driving internalization of tissues during animal development[18]. A large body of experimental evidence indicates that apical constriction driven by phosphorylation and activation of the molecular motor myosin II is required for tissue invagination[6]. However, the extent to which apical constriction is on its own sufficient to drive tissue internalization is unknown. During this process, cells undergo a series of complex changes in shape and intracellular organization, whose causal relationship to apical constriction and inward folding remain poorly understood[19–22]. Furthermore, the organismal scale cells occupy defined positions and are organized in specific geometrical patterns, which might facilitate or constrain invagination. Finally, apical constriction is not always coupled with invagination and several invagination processes are independent of apical constrictions[23]. For example, during salivary gland invagination, apical constriction and tissue invagination are uncoupled. When apical constriction is inhibited, compressing forces exerted by a supracellular myosin cable surrounding the salivary gland pit are sufficient to push cells inward[24]. Similar actomyosin-cable-mediated forces drive neural tube closure during chick embryogenesis[25]. Other examples of invaginations independent of apical constriction include the folding of Drosophila leg epithelium, which is driven by whole-cell shrinkage coupled with apoptosis[22], and ascidian gastrulation, which is driven by a basolateral accumulation of myosin II and apicobasal cell shortening[26]. In addition, basal wedging rather than apical constriction seems to be the major force driving tissue internalization during mouse neuronal tube development[27]. Even in the case of Drosophila ventral furrow invagination, arguably the best characterized example of epithelial folding, the extent to which apical constriction drive invagination is unknown. Computer simulations suggest the requirement of additional pushing forces exerted by lateral ectodermal cells[28,29], and relaxation of the basal surface of invaginating cells[30]. At the tissue-scale, the emergence of collective contractile behavior and its relationship to tissue geometry and invagination also remains the focus of active investigations[31–33]. In this study, we employ an optogenetic method to synthetically reconstruct epithelial folding during early Drosophila embryogenesis. In this context, "synthetic" refers to guided spatio-temporal control over the signaling pathway driving apical constriction, which is otherwise dependent upon the differentiation program of the embryo. Using this approach, we test the extent to which apical constriction on its own can drive invagination, and how different contractile behaviors arise in response to different temporal and spatial patterns of optogenetic activation. Collectively, our results indicate that apical constriction is sufficient to drive tissue invagination, but it is not sufficient to fold an invagination into a tube-like shape. Furthermore, our results provide insights into the emergence of pulsatile contractions and impact of tissue geometry on coordinated contractile behavior.

## Results

**RhoGEF2 plasma membrane recruitment and tissue responses**. To study the impact of apical constriction on tissue folding, independently of any pre-defined conditions, associated with normal invagination processes, we used an optogenetic system to activate Rho signaling[34,35] at the apical surface of developing Drosophila embryos prior to any sign of morphological differentiation (Fig. 1a, b). At the end of cellularization, the Drosophila embryo is composed of a monolayer of epithelial cells without any morphological difference along the antero-posterior (AP) or dorso-ventral (DV) axes. To avoid any contribution from endogenous invagination processes, experiments were performed on the dorsal side of the embryo. In this region, the twist and snail transcription factors, that control ventral furrow invagination[36], are not expressed and there is no morphological movement at this stage[37].

Apical constriction is initiated by activation of the small GTPase Rho1 by exchange factors of the RhoGEF family, which results in myosin activation and contraction of cortical actin filaments[38,39]. We employed the CRY2/CIB1 protein heterodimerization system[32,40] to control the plasma membrane localization of Drosophila RhoGEF2 using light. We engineered embryos to co-express RhoGEF2-CRY2 (the catalytic domain of RhoGEF2 fused to CRY2, a light-sensitive protein domain) and CIBN::pmGFP (consisting of the N-terminal domain of the CRY2 binding partner CIB1 tagged with a plasma membrane anchor and GFP), see cartoon in Fig. 1a. In the dark, embryos developed normally, RhoGEF2-CRY2 localized to the cytoplasm, and CIBN:: pmGFP was correctly anchored at the plasma membrane (Fig. 1c, f, i). Photo-activation using a two-photon based protocol ($\lambda = 950$ nm)[41] caused recruitment of RhoGEF2-CRY2 to the plasma membrane within 30 s (Fig. 1d, g, i), and the moving away of cells (within ~ 5 min) from the imaging plane in an area that precisely matched the geometry (circular, triangular, or squared) of the illuminated area (Fig. 1e, h, k and Supplementary Movie 1).

**RhoGEF2 optogenetic activation causes apical constriction**. In order to test whether optogenetic activation caused apical constriction and at the same time visualize both activated and non-activated cells, we imaged embryos co-expressing RhoGEF2-CRY2 and CIBN::pmGFP, together with the plasma membrane marker Gap43 tagged with mCherry (Gap43::mCherry). We photo-activated a rectangular area on the dorsal surface of the embryo (blue box in Fig. 2a) and followed the cell surface area over time using quantitative imaging (Fig. 2f–i and Supplementary Movie 2). Cells contained within the box of activation constricted their apical area up to ~60% with a 20% increase in eccentricity (Fig. 2c–e, g), while they ingressed towards the interior of the embryo within ~6 min (Fig. 2b, c). Consistently, myosin II accumulated specifically at the apical surface of activated cells (Fig. 2j–r and Supplementary Movie 3). Non-activated cells, located at the boundary between the activated and non-activated regions, were pulled towards the activated region and increased their apical area by 50% (Fig. 2b–g). Non-activated cells outside of the boundary region did not change their apical surface significantly over time (Fig. 2b–f). TUNEL assay revealed that photo-activation did not induce apoptosis (Supplementary Fig. 1a, c, e), and in agreement with previous reports, apoptosis could only be detected at later stages of embryogenesis[42] (Supplementary Fig. 1b, d, f). Thus, recruitment of RhoGEF2 to the plasma membrane is on its own sufficient to cause apical constriction and tissue-level responses that are compatible with endogenous invagination processes.

**RhoGEF2 optogenetic activation and pulsatile constrictions**. Apical constriction is often characterized by pulsatile cycles of

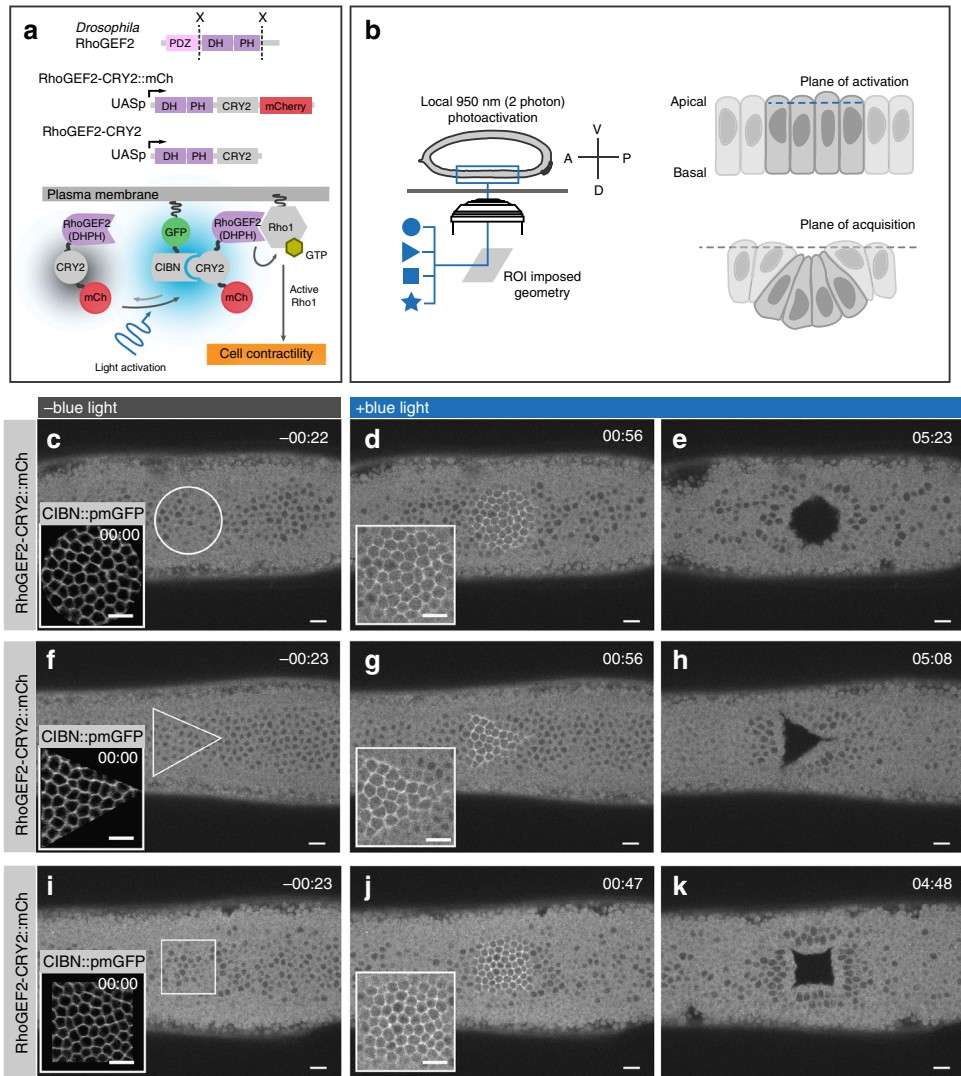

**Fig. 1** Light-mediated recruitment of RhoGEF2 at the plasma membrane causes tissue-level responses. **a** Schematic illustration of the CRY2/CIBN optogenetic system used for controlling plasma membrane recruitment of the Rho1 activator RhoGEF2 during early *Drosophila* embryogenesis. The DHPH catalytic domain of the GTP Exchange factor RhoGEF2 was fused to the light-sensitive PHR domain of CRY2 and either tagged with mCherry or left without a fluorescent tag. In the absence of light, RhoGEF2-CRY2::mCherry localizes to the cytoplasm (left). Upon two-photon illumination, CRY2 undergoes a specific conformational change and binds to the plasma membrane anchor CIBN::pmGFP (right). At the plasma membrane, RhoGEF2-CRY2::mCherry triggers activation of Rho1 and its downstream effectors. **b** Cartoon illustrating the experimental setup used in this study. Stage 5 *Drosophila melanogaster* embryos expressing the optogenetic module were mounted with their dorsal epithelium facing the coverslip, and a region of interest (ROI) of the desired geometrical pattern (left) was defined to delimit the area of two-photon illumination. The plane of activation was restricted to the junctional plane at 4 μm from the apical side of the epithelium (right top); over time photo-activated cells disappeared from the plane of acquisition (right bottom). **c**–**k** Confocal still frames from time-lapse recordings of three representative embryos ($N = 9$) co-expressing CIBN::pmGFP and RhoGEF2-CRY2::mCherry. Images are presented as integrated intensity projections of 3 μm. **c** In the absence of blue-light (~20 s before photo-activation), RhoGEF2-CRY2::mCherry localizes to the cytoplasm. Inset shows the plasma membrane localization of CIBN::pmGFP at the time of photo-activation using two-photon illumination. **d** RhoGEF2-CRY2::mCherry recruitment to the plasma membrane was restricted to the imposed circular area after eight consecutive rounds of photo-activation for a total time of 2 s. In depth, the area of activation was restricted to three consecutive z-stacks centered at 4 μm from the apices of the cells. Inset represents a magnification of the activated area. **e** Sustained photo-activation was alternated with mCherry excitation and recorded in intervals of ~21 s. After ~5 min, a hollow matching the geometrical pattern of photo-activation formed on the dorsal epithelium as photo-activated cells disappeared from the plane of acquisition. Similar experiments were carried out with a triangle (**f**–**h**) and a square (**i**–**k**). Scale bars are 10 μm

contraction and expansion of the surface area, as seen during dorsal closure[43], or of contraction and stabilization, as seen during ventral furrow formation[44]. The apical constrictions induced by optogenetic stimulation of RhoGEF2 displayed a monotonic increase and decrease in the constriction rate over a period of ~8 min (Fig. 2h) without showing any obvious pulsatile behavior in a cell-by-cell analysis of the constriction rate (Fig. 2i), thus arguing that the average contractile behavior did not obscure any underlying individual pulsatile behavior. Our initial photo-

activation protocol was based on a continuous and synchronous administration of light, which might override any potential pulsatile behavior induced by endogenous biochemical or mechanical processes downstream of Rho signaling. Therefore, we monitored apical constriction under different illumination conditions. First, we tested whether changing the frequency of illumination without modifying the laser power (10 mW) would be sufficient to elicit pulsatile constrictions. The results from these experiments demonstrated that, at intervals of ~55 s of

illumination, cells displayed a pulsatile behavior (Fig. 3a and Supplementary Movie 4), whereas at intervals below and above this value, cells either did not constrict or constricted continuously and invaginated. At 55 s intervals, activated cells displayed cycles of apical constriction and expansion with a mean period of $157 \pm 63$ s (Fig. 3d), that lasted for about 10 min before endogenous gastrulation movements caused cells to move away from the photo-activation plane, making them untrackable for a longer period of time. This contractile behavior was not sufficient to cause tissue internalization but rather resembled the pulsatile

behavior of amnioserosa cells during dorsal closure or that of ventral cells in *twist* or *fog* mutant embryos, which lack the stabilization phase and fail to invaginate[44]. However, differently from these two conditions, constrictions induced by pulsed illumination were synchronized. Cells constricted and relaxed in phase as demonstrated by the preservation of oscillatory behavior even after averaging out time-course measurements of multiple cells (Fig. 3a–c). Considering the reversion kinetics of the CRY2/CIB1 system ($T_{1/2} \sim 5$ min)[32,40], it is unlikely that an administration of light at 55 s intervals was directly inducing contraction

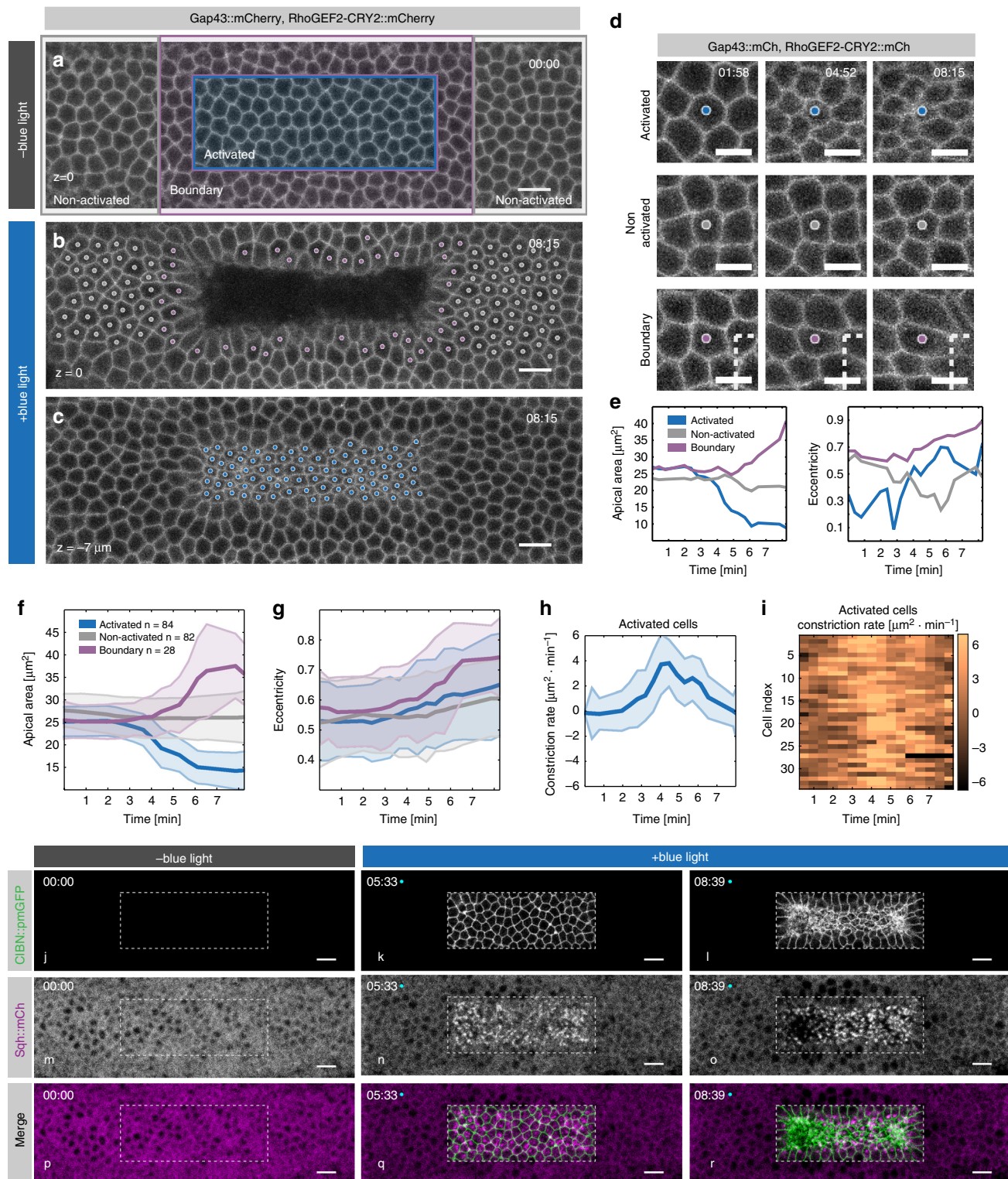

and relaxation at each light on/off cycle. Consistently, when the time-point of illumination was plotted over the time-course analysis of the surface area, no direct correlation between contraction and illumination could be observed (Fig. 3e). This suggests that this discontinuous illumination regime triggers an endogenous mechanical and/or chemical oscillatory system whose dynamics were probably altered by the continuous illumination protocol previously employed. Therefore, we tested whether either a continuous administration of light at a lower laser power or a single pulse of light at a higher laser power would also elicit pulsatile behavior. Continuous illumination at 5 mW induced cycles of contraction and relaxation with a mean period of 148 ± 41 s (Fig. 3b, d and Supplementary Movie 4). Similarly, a single pulse of light at a higher laser power (15 mW) also induced pulsatile behavior, albeit with a slightly slower mean period (176 ± 58 s) (Fig. 3c, d and Supplementary Movie 4). Taken together, these data are compatible with the presence of an endogenous mechano-chemical oscillatory system[45] that can be induced by Rho signaling activation up to a certain threshold, above which cells contract continuously without pulsing.

Next, we established a photo-activation protocol that allowed us to both induce pulsatile contractions and, at the same time, visualize cell shape and myosin II dynamics. A single pulse of two-photon optogenetic activation (15 mW) was followed by continuous illumination at a lower laser power (2 mW), which allowed the imaging of the apical surface (CIBN::pmGFP) at high temporal resolution without causing inhibition of pulsations. Myosin II labeled with mCherry (Sqh::mCherry) was visualized by alternating two-photon illumination with single photon illumination (561 nm) at 13 s intervals (Fig. 4a–f and Supplementary Movie 5). Under this condition, cells pulsed with a mean period of 164 ± 49 s (Fig. 4c) with myosin II pulsing with similar kinetics (Fig. 4b, mean period 159 ± 52 s). Myosin II accumulated at the medio-apical plane as cells contracted and then gradually disappeared as cells relaxed (Fig. 4d–f). Quantitative analysis revealed a temporal correlation between myosin II pulses and apical constriction with ~40% of constrictions occurring within 25 s of a myosin pulse (Fig. 4h), and with the average maximum correlation coefficient occurring when the myosin II accumulation rate was not shifted compared to the apical constriction rate (Fig. 4g). These results argue that optogenetic-induced pulsatile behavior arises as a consequence of synchronous myosin medio-apical accumulation and dissolution.

**Continuous RhoGEF2 activation and deep tissue invagination.**
The data so far collected suggest a direct relationship between strength of Rho signaling activation and tissue invagination. To further test the relationship between signal strength and invagination, we assessed whether continuous activation of Rho signaling would be sufficient to drive complete invagination. Cells were photo-activated at the apical surface of the embryo and the photo-activation plane was adjusted throughout the course of the experiment to follow the invaginating cells as they moved away from the focal plane of imaging. A series of z-stacks covering the entire apical–basal length of the cells was acquired at different time points and projected on a zy-plane. After 10 min of photo-activation, cells that moved away from the surface of the embryo formed a shallow indentation (Fig. 5a–c), which rapidly (<1 min) acquired a more convex form (Fig. 5d–f) before folding into a U-shaped invagination (Fig. 5g–i). Imaging of RhoGEF2-CRY2 and myosin II confirmed selective optogenetic activation at the apical surface, with both markers displaying a maximal spreading in z along the apicobasal axis of the cells of ~4 microns (Fig. 6a–f). Similar optogenetic-guided invaginations could also be induced at later stages of embryogenesis, when the embryo had already acquired a more complex morphology (Supplementary Fig. 2), arguing that recruitment of RhoGEF2 to the plasma membrane is on its own sufficient to cause tissue invagination on a time-scale compatible with endogenous invagination processes.

**Tissue geometry and the orientation of cell contractions.**
During endogenous invagination processes, cells are arranged in predefined geometrical patterns, which seem to impact on apical contractile behavior. For example, during ventral furrow formation, cells constrict preferentially along the DV axis and elongate along the AP axis (AP anisotropic constriction)[46]. This contractile behavior correlates with the geometry of the ventral furrow primordium, which is composed of cells organized in a rectangular shape oriented with its major axis parallel to the AP axis of the embryo[32]. However, ventral cells might also be subject to additional forces and constraints imposed, for example, by the specific curvature of the embryo or by its overall oblong morphology. To test whether there is a direct link between tissue geometry and contractile behavior, we monitored apical constriction in synthetic furrows of different quadrilateral shapes and orientation (Fig. 7 and Supplementary Movie 6). Cells arranged in a squared geometry constricted isotropically (Fig. 7c, d), while cells arranged in a rectangular pattern constricted anisotropically along an axis perpendicular to the major edge of the rectangular pattern that was designed and thus acquired an elongated shape along that edge (Fig. 7k, l). The degree of anisotropy increased as a function of the rectangularity of the furrow: the higher the ratio between the major and the minor edge, the higher the degree of anisotropy (Fig. 7e–m). Changing the orientation of a synthetic furrow by 90° (i.e., a furrow that was designed with its major axis perpendicular to the AP axis of the embryo) caused a

**Fig. 2** RhoGEF2 optogenetic activation triggers apical constriction. **a–d** Confocal still frames from a time-lapse recording of a representative embryo ($N = 3$) co-expressing CIBN::pmGFP, RhoGEF2-CRY2::mCherry, and the plasma membrane marker Gap43::mCherry. **a** The dorsal epithelium prior to two-photon illumination as visualized by imaging Gap43::mCherry, note that with the imaging setting used, the cytoplasmic signal of RhoGEF2-CRY2::mCherry was not visible. Blue rectangle indicates area of photo-activation. Cells within the magenta rectangle were defined as boundary cells. Cells contained within the gray and magenta rectangle corresponded to non-activated cells. **b** and **c** show final still frames after 8 min of continuous two-photon illumination. **b** shows the apical-most plane. **c** shows a focal plane 10 μm basally to the focal plane in **b**. Individual cells marked with a dot (color-coded as indicated above) were tracked over-time and their apical area was quantified, as presented in **f**. Scale bars, 10 μm. **d, e** Detailed analysis of individual cells from the embryo shown in **a**. **d** Cell outlines of activated (top row), non-activated (middle row), and boundary (bottom row) cells after photo-activation; dashed line corresponds to the top left corner of the area of photo-activation. Scale bars are 5 μm. **e** Quantification of apical area (left) and eccentricity (right) over time for the individual cells shown in **d**. **f** Quantification of mean apical area over time upon two-photon illumination of activated ($n = 84$), non-activated ($n = 82$), and boundary cells ($n = 28$). Shaded areas indicate s.d. **g** Mean apical area eccentricity quantified for the same groups of cells as in **f**. Shaded areas indicate s.d. **h** Quantification of mean apical constriction rate over time for the photo-activated cells in **f** ($n = 84$). Shaded areas indicate s.d. **i** Quantification of the constriction rate of individual cells within the photo-activated region ($n = 34$). **j–r** Confocal still frames from a time-lapse recording of a representative embryo ($N = 3$) co-expressing CIBN::pmGFP, RhoGEF2-CRY2, and the myosin II regulatory light chain reporter Sqh::mCherry. Images are presented as integrated intensity projections of 5 μm. The dorsal epithelium was subjected to continuous two-photon illumination within a rectangular area (dashed line). **j–l** CIBN::pmGFP, (**m–o**) Sqh::mCh, and (**p–r**) composite of both (merge)

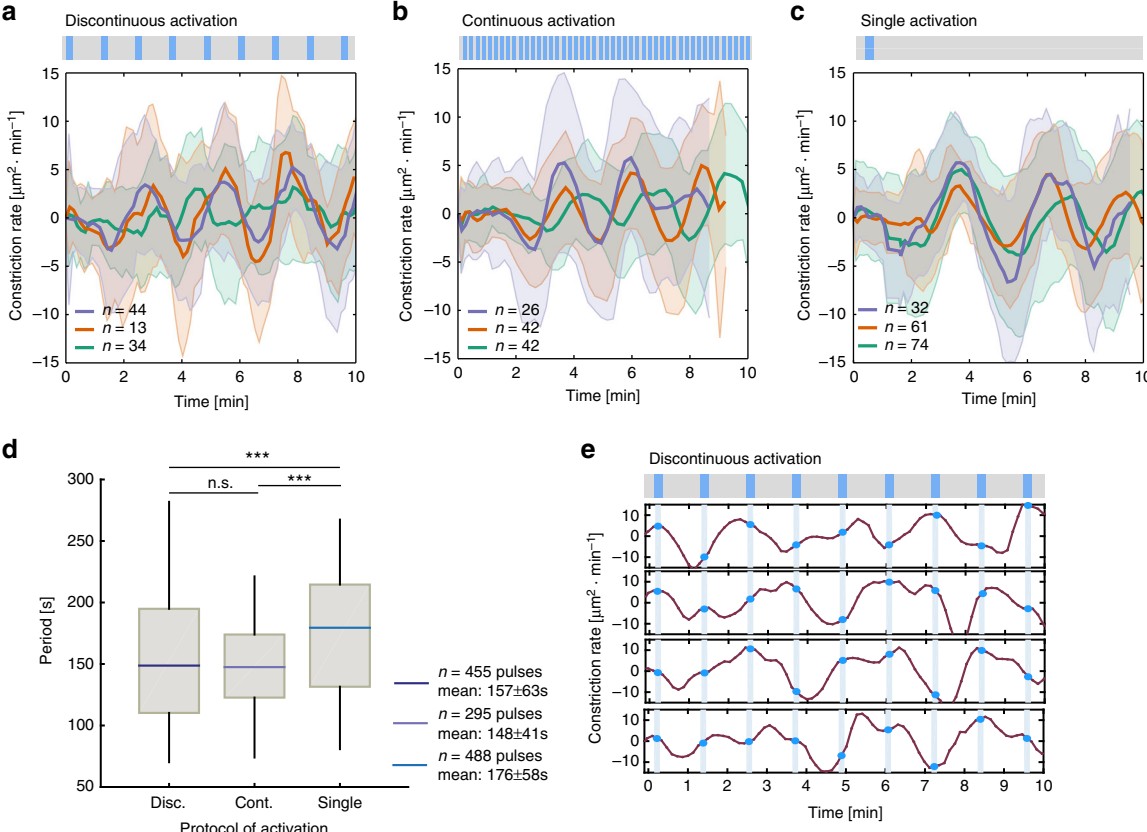

**Fig. 3** Induction of pulsatile apical contractions. **a–c** Constriction rate of photo-activated cells measured by change in apical area over time. Positive values indicate contractions while negative values indicate expansions. Embryos co-expressing CIBN::pmGFP, RhoGEF2-CRY2::mCherry, and the plasma membrane marker Gap43::mCherry (N = 9) were subjected to different illumination protocols. In all cases, cell outlines were recorded in mCherry (Gap43:: mCherry) during and after photo-activation. Data presented as mean (solid line) and s.d. (shaded area). A rectangular region on the dorsal epithelium was photo-activated using two-photon illumination in a single plane, centered at 4 μm from the apical-most plane. **a** Discontinuous photo-activation (10 mW) with a resting interval of 55 s (N = 3 embryos, n = 44, 13, 34 cells). **b** Continuous photo-activation (5 mW, N = 3 embryos, n = 26, 42, 42 cells). **c** Single initial photo-activation (15 mW, N = 3 embryos, n = 32, 61, 74, cells). **d** Box-and-whisker plot of the quantified period between peaks of constriction for the indicated protocols. The quantified constriction pulses correspond to the data in **a–c**, (n = 455, 295, 488 pulses, ***p < 0.001, two-tailed paired t-test). **e** Constriction rate over time for four individual cells during the discontinuous activation protocol (blue markers indicate time of illumination)

re-orientation of the anisotropy, with cells constricting pre-ferentially along the AP axis and elongating along the DV axis (Fig. 7a, b, n). Thus, these data demonstrate a direct correlation between anisotropic constriction and tissue geometry, independently of the overall shape of the embryo.

## Discussion

Taken together the results presented in this study show that localized activation of Rho signaling at the apical surface of cells, which are otherwise not programmed to invaginate, is sufficient to cause tissue invagination and to recapitulate major cell- and tissue-level behaviors associated with endogenous invagination processes. As discussed in detail in the Introduction, mechanisms other than apical constriction control a variety of different forms of invaginations during animal development. Our results do not challenge this view, rather, they argue that if considering a monolayer of epithelial cells, apical constriction is sufficient to fold it into a U-shape invagination. However, apical constriction is not sufficient to drive closure of an invagination into a tube-like structure, as seen for example during ventral furrow formation. Additional pushing forces exerted by lateral non-invaginating cells and/or loss of myosin II from the basal surface and basal relaxation might be required to complete this step, as suggested by a recent study based on a 2D finite element model[30].

At the tissue-level, the contractile behavior of individual cells depends on the geometry of photo-activation. While a square box results in isotropic apical constrictions, a rectangular shape causes cells to constrict preferentially along the minor axis of the photo-activated area and to elongate along the major axis. This anisotropic contractile behavior resembles the one of ventral furrow cells, which are also organized in a rectangular pattern and constrict preferentially along the short axis of the tissue. Anisotropy in ventral furrow cells is not genetically determined but arises as a consequence of tissue geometrical and mechanical constraints[32,33]. Consistent with these studies, the increase in the degree of anisotropic constriction as a function of the rectangularity of the photo-activated area presented in Fig. 7 can be explained if considering that it is mechanically less favorable to shrink cells along the major axis of a rectangle than along the short axis. Indeed, the former deformation requires the endpoints of the constricting tissue to move farther, and thus a larger deformation of neighboring tissues along that axis.

Our results also reveal an interesting correlation between pulsatile constrictions and tissue invagination. During endogenous morphogenetic processes, two different pulsatile behaviors have been described. One is based on cycles of myosin II accumulation at the medio-apical plane of the cell, during the contraction phase, and dissolution during the relaxation phase. This type of pulsatile behavior has been first described during dorsal

closure in *Drosophila* and it is not linked to tissue invagination[43]. Another type of pulsatile behavior is based on an incremental accumulation of myosin II at each contraction, which is followed by a stabilization period of cell shape without an intervening relaxation phase (ratcheted contractions)[44]. Ratcheted contractions require the radial polarization of Rho signaling components. However, in certain mutant conditions that interfere with the establishment of radial polarity, contractions become non-ratcheted with myosin II miss-localizing at three-ways junctional vertices[39]. The optogenetic-induced pulsatile contractions described in Fig. 3 display a non-ratchet behavior, and similarly to dorsal closure contractions, are characterized by myosin II medio-apical accumulation and dissolution in phase with contraction and relaxation of the apical surface. However, differently from dorsal closure, optogenetic-induced pulsatile contractions display a higher degree of synchrony with photo-activated cells constricting and relaxing in concert. This difference could be explained if considering that a light pulse provides a coherent and synchronous input, while activation of signaling in a developing tissue might be more subject to cell-to-cell variability. Lack of

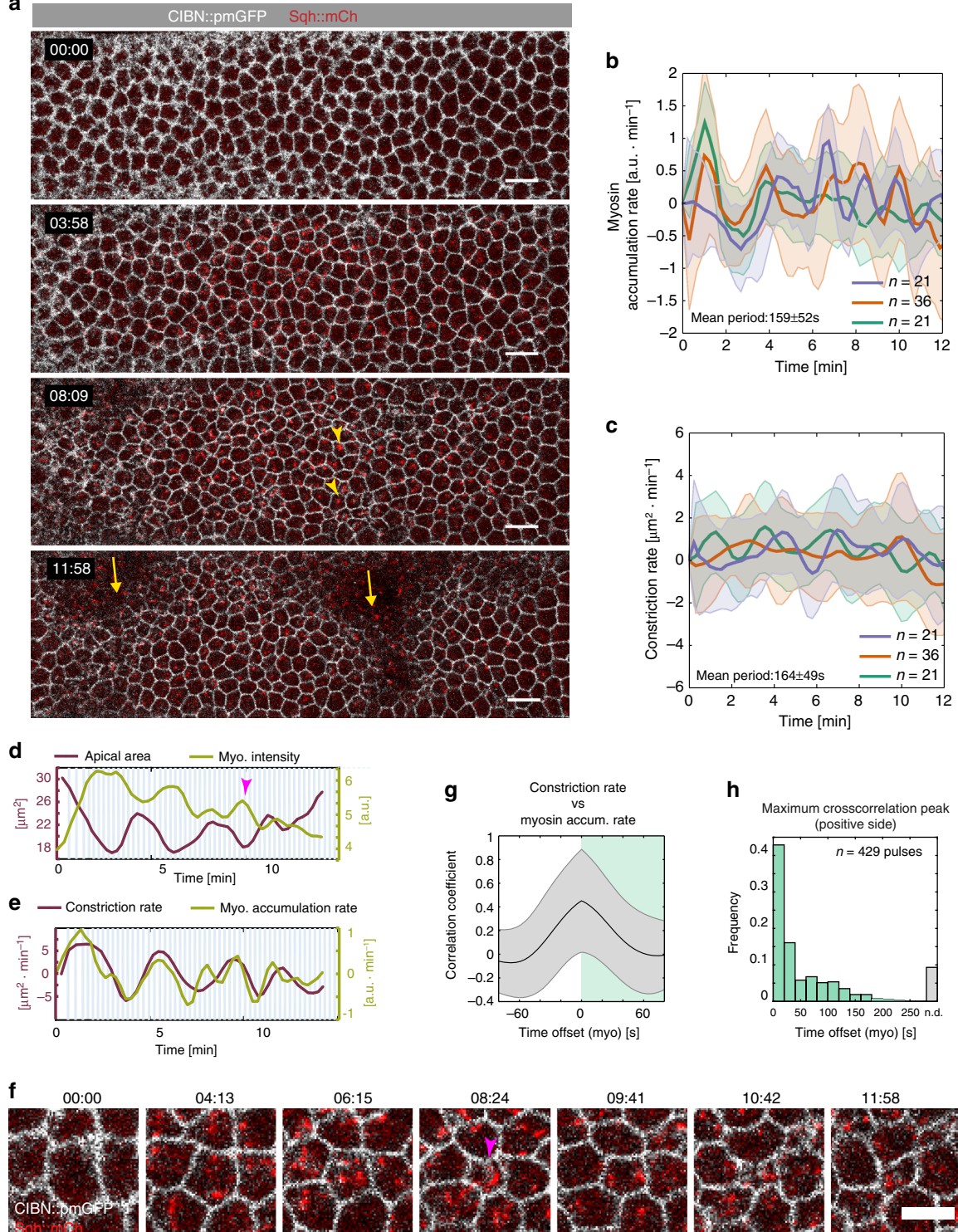

tissue internalization upon induction of pulsatile constrictions is likely due to the absence of a stabilization phase after constriction of the apical surface, which might result in a dissipation of the forces that are normally needed to build tension and drive invagination. Consistently, continuous administration of light induced synchronous contractile behavior and invagination, mimicking the activity of signaling molecules such as *fog* whose function is to control the transition from stochastic to collective contractile behavior during ventral furrow invagination[47]. Pulsatile behavior could be elicited either by a discontinuous administration of light, or by continuous illumination at a lower laser power, or by a single pulse at a higher laser power. We interpret these results to suggest that pulsations can be induced by the stimulation of a Rho-dependent mechano-chemical oscillatory system up to a certain threshold, above which cells constrict without pulsing. Stimulation of Rho signaling above a certain threshold could override, for example, the activity of a RhoGAP, which is required to control the normal spatiotemporal dynamics of Rho GDP/GTP cycling. In agreement with this proposal, pulsatile constrictions during ventral furrow invagination require the activity of a specific RhoGAP[48]. However, while ventral cells pulse with a mean period of ~80 s[44], optogenetic-induced pulsations display a mean period of ~150 s, a limit probably imposed by the reversion kinetics of the CRY2/CIB1 system in the dark[32,40].

In conclusion, these data illustrate the utility of applying concepts of synthetic biology (e.g., precise orthogonal control over signaling pathways, guided cell behavior) to the field of tissue morphogenesis and in particular of how the nascent field of synthetic morphogenesis can help defining the minimum set of requirements sufficient to drive tissue remodeling. Our data argue that while normally tissue differentiation and tissue shape are intimately linked, it is possible to direct tissue shape without interfering with complex layers of gene regulatory network and tissue differentiation programs. This might have important implications also for tissue engineering, where it might be desirable to shape any given tissue of interest without changing its fate.

## Methods

**Cloning**. To generate RhoGEF2-CRY2::mCherry, the DHPH catalytic domain of RhoGEF2 was PCR-amplified from *Drosophila melanogaster* cDNA using gene-specific primers and joined with the CRY2::mCherry sequence through overlap extension PCR (aa 1177–1554, from the RhoGEF2 reference sequence NP_995869.1). Non-fluorescently-tagged RhoGEF2-CRY2 was cloned by amplifying RhoGEF2-CRY2 from RhoGEF2-CRY2::mCherry. Both constructs were cloned into the pPW vector (Drosophila Genomics Resource Center, Bloomington, IN) using the Gateway cloning system (Life Technologies) according to standard procedures.

Because the available UAS-sqh-Gap43::mCherry plasma membrane marker harboring UAS regulatory sequences caused morphological abnormalities when expressed at high level in combination with Gal4 expression; we generated a spaghetti-squash-promoter-only driven Gap43::mCherry (sqhp>Gap43::mCherry). To generate this construct, both the *D. melanogaster* spaghetti squash promoter gene sequence (−1175 to +153 from the Sqh reference sequence NM_001298002) and mCherry were amplified with gene-specific primers that additionally contained the Gap43 sequence. The fragments were then joined together into pCasper5, cut with NotI and AgeI, using Gibson Assembly.

**Fly strains and genetics**. The following transgenic fly lines were obtained by standard methods and all stocks were maintained at 22 °C. Standard genetic crosses were kept in the dark and were used to generate flies having the genotypes described below.

To show the effect RhoGEF2-CRY2 plasma membrane recruitment on the dorsal epithelium:

w[*]; P[w+, UASp>CIBN::pmGFP]/+; P[w+, UASp>RhoGEF2-CRY2::mCherry]/P[w+, Osk>Gal4::VP16].

To follow cell outlines upon RhoGEF2-CRY2 activation:

P[w+, sqhp-Gap43::mCherry]/w[*]; P[w+, UASp>CIBN::pmGFP]/+; P[w+, UASp>RhoGEF2-CRY2::mCherry]/ P[w+, Osk>Gal4::VP16].

To visualize myosin II dynamics upon RhoGEF2-CRY2 activation:

w[*]; P[w+, UASp>CIBN::pmGFP]/P[w+, Sqh::mCherry]; P[w+, UASp>RhoGEF2-CRY2]/ P[w+, Osk>Gal4::VP16].

To visualize both membranes and nuclei together:

w[*]; P[w+, UASp>CIBN::pmGFP]/P[w+, his2Av::mRFP1]; P[w+, UASp>RhoGEF2-CRY2]/ P[w+, Osk>Gal4::VP16].

For embryo collection, flies of the desired genotype were collected into cages with apple juice plates and yeast paste. All cages were kept in the dark at 22 °C.

**Fly stocks**. w[*]; If/CyO; P[w+, UASp>RhoGEF2-CRY2::mCherry]/TM3, Ser (this study). UASp-Gal4-driven expression of the catalytic DHPH domain of RhoGEF2 fused upstream of CRY2 PHR domain and C-terminally tagged with mCherry fluorescent protein.

w[*]; If/CyO; P[w+, UASp>RhoGEF2-CRY2]/TM3, Ser (this study). UAS-Gal4-driven expression of the catalytic DHPH domain of RhoGEF2 fused upstream of CRY2 PHR domain.

P[w+, sqhp>Gap43::mCherry]/Fm7; Sb/TM6 Tb (this study). Gap43 membrane marker fused to mCherry fluorescent protein in which expression is driven only by the Spaghetti-squash promoter.

w[*]; P[w+, UASp>CIBN::pmGFP]/Cyo; Sb/TM3, Ser. Membrane-anchored CIBN additionally fused to EGFP.

w[*]; If/CyO; P[w+, his2Av::mRFP1]/TM3, Ser. mRFP1 tagged Histone 2A version A. (Bloomington stock number 23651.)

w[*]; P[w+, Sqh::mCherry]/CyO; Sb/TM3, Ser. Myosin regulatory light chain Spaghetti-squash tagged with mCherry fluorescent protein.

w[*]; If/CyO; P[w+, Oskp>Gal4::VP16]/TM3, Ser. Maternally deposited Gal4 protein driven by the Oskar promoter. (Bloomington stock number 44242.)

**Live imaging and optogenetics**. For collection of light-sensitive embryos, cages were kept and manipulated in the dark. Stage 5 embryos were identified and mounted using a standard stereomicroscope under transmitted illumination. To prevent unwanted photo-activation, the microscope light source was replaced with a conventional red-emitting LED lamp. Following selection under halocarbon oil, the embryos were dechorionated with 90% sodium hypochlorite for 2 min, washed with water and mounted with PBS onto a 35 mm glass-bottom dish (MatTek corporation). The embryos were positioned with their dorsal side facing the coverslip.

**Fig. 4** Medio-apical myosin II accumulation and pulsatile changes in cell shape. **a** Merged confocal still frames from a time-lapse recording of a representative embryo ($N = 3$) co-expressing CIBN::pmGFP (cell outline, in white), RhoGEF2::CRY2 (not visible), and the myosin II regulatory light chain reporter Sqh::mCherry in red (yellow arrowheads). Images are presented as maximum intensity projections (5 µm) from the apical-most plane. Cells were subjected to an initial two-photon illumination pulse (15 mW) followed by a continuous two-photon excitation at a lower laser power (2 mW) for image acquisition. Imaging was continued for 12 min until formation of cephalic furrow and posterior dorsal folds started to displace the tissue (left and right yellow arrows, respectively). Scale bars, 10 µm. **b** Rate of myosin II accumulation over time quantified from integrated intensity projections of Sqh::mCherry signal within 5 µm of the cells' apical-most plane (three embryos, with $n = 21, 36$, and 21, cells respectively). Results are presented as mean and s.d. **c** Corresponding changes in cell-shape over time quantified by the rate of constriction for the same three embryos as in **b**. Results presented as mean and s.d. **d**, **e** Quantification of apical area and myosin II intensity (**d**) and corresponding constriction and myosin II accumulation rates (**e**) for an individual cell shown in **f**. The arrowhead in magenta indicates an example of a contractile event. **f** Detailed view of an individual cell as it undergoes cycles of contraction and expansion as quantified in **d** and **e**. Magenta arrowhead indicates a contractile event. Scale bar is 5 µm. **g** Correlation coefficient resulting from temporally shifting myosin II accumulation rate with respect to the apical constriction rate (429 pulses in 78 cells from three embryos) presented as mean and s.d. The average maximum correlation coefficient occurs when myosin accumulation rate is not shifted. **h** Distribution of time offsets at a peak of maximum correlation for individual cross-correlation curves calculated in **g**. 40% of apical contractions occurred within a 20 s interval from a myosin peak (first bin in the graph). n.d. indicates that no peak of maximum correlation was found between two consecutive contractile events (~10%)

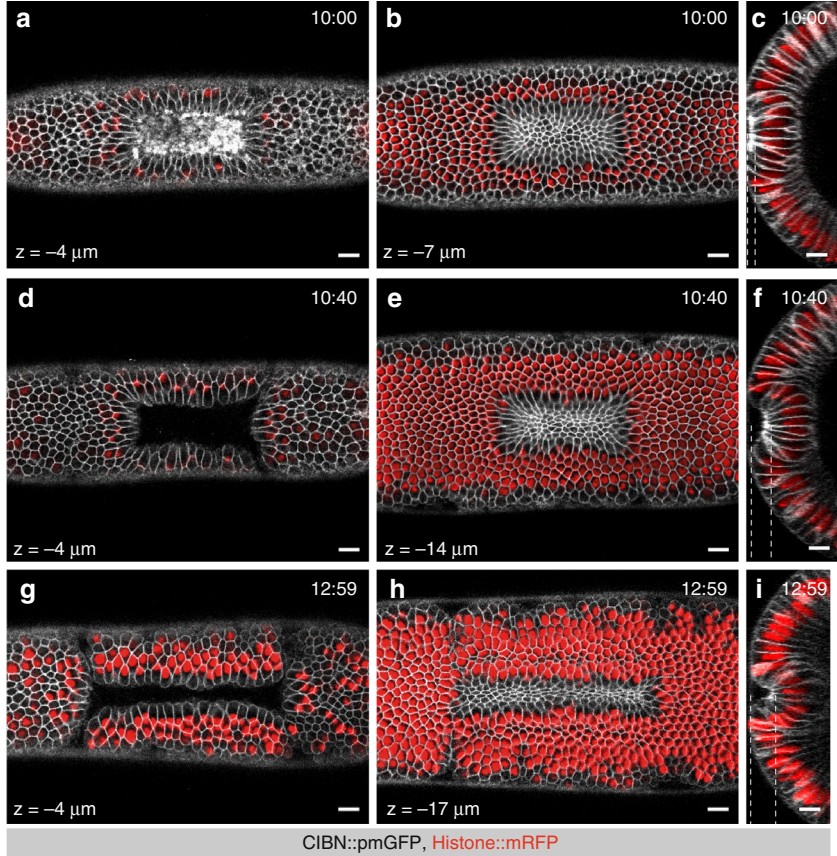

**Fig. 5** Sustained RhoGEF2 optogenetic activation is sufficient to drive a complete invagination. **a–i** Confocal images of three individual embryos co-expressing CIBN::pmGFP, RhoGEF2-CRY2, and Hist2A::mRFP as nuclear marker. Within a rectangular on the dorsal epithelium, two-photon illumination was restricted to three consecutive z-stacks (1 μm spacing) centered at 4 μm from the apical-most plane, and excited for ~152 ms during each acquisition round. Sustained photo-activation was alternated with mCherry excitation at intervals of ~13 s. As the epithelium folded inwards, continuous two-photo-activation was achieved by manually adjusting the ROI such that illumination was restricted to the initial group of photo-activated cells. In addition, the focal plane of activation was manually rectified to match the cells' apical-most plane. At the indicated time points, a z-stack comprising the dorsal epithelium was acquired for both mRFP and GFP (50 planes, 1 μm spacing). Overlays of both channels are shown. **a–c** Three views of the dorsal epithelium after 10:00 min of continuous activation. **a** Surface view of the dorsal epithelium at a depth of 4 μm. **b** Apices of activated cells at a depth of 8 μm. **c** Cross-section view of the activated area. **d–f** Three views of the dorsal epithelium after 10:48 min of continuous activation. **d** Surface view of the dorsal epithelium at a depth of 4 μm. **e** Apices of activated cells at a depth of 14 μm. **f** Cross-section view of the activated area. **g–i** Three views of the dorsal epithelium after 12:59 min of continuous activation. **g** Surface view of the dorsal epithelium at a depth of 4 μm. **h** Apices of activated cells at a depth of 17 μm. **i** Cross-section view of the activated area. Dashed lines indicate the depths at which surface views were acquired (left and middle panels). Scale bars, 10 μm

Live imaging and photo-activation experiments, as well as image acquisition of TUNEL and antibody staining, were carried out at 20 °C with a Zeiss LSM 780 NLO confocal microscope (Carl Zeiss) equipped with a 561 diode laser, an argon laser, and a tunable two-photon laser (690–1040 nm) (Chameleon; Coherent, Inc.). A 40×/NA 1.1 water immersion objective (Carl Zeiss) was used for image acquisition. For sample location, bright field illumination was filtered through a Deep Amber lighting filter (Cabledelight, Ltd.). The microscope was controlled through the Zen Black software (Carl Zeiss) whereas photo-activation protocols were carried out with the Pipeline Constructor Macro[49]. For all experiments, an initial mCherry frame was acquired prior to activation (acquisition at 561 nm) and followed by time-lapse recording of alternating mCherry and local two-photon excitations. For experiments in which transversal cross-sections are shown, a z-stack comprising the dorsal epithelium was acquired (69 planes, 1 μm spacing) in mCherry during photo-activation and in GFP posterior to photo-activation. Local two-photon photo-activation was achieved at 950 nm with laser power set to 10 mW and scanning direction set to bidirectional yielding a pixel dwell of 1.27 μs. Unless otherwise stated, these settings were maintained throughout experiments. Laser power measurements are reported as measured at 1 cm from the objective.

For assessing the effect of RhoGEF2-CRY2::mCherry plasma-membrane recruitment, the dorsal epithelium was activated within the given ROI (circle, triangle, square) and limited to three consecutive z-stacks (1 μm spacing) centered at 4 μm from the apical-most plane. An initial mCherry stack was acquired followed by eight consecutive rounds of two-photon activation for a total effective illumination time of 2 s (~257 ms per illumination round at 10 mW laser

power). Subsequently, a time-lapse recording of alternated mCherry and local two-photon excitation was acquired. The time between two-photon excitations was 21 s.

In order to follow apical constriction, myosin II dynamics, and any potential induction of apoptosis upon photo-activation, a ROI of activation on the dorsal epithelium was designed. An initial mCherry frame was acquired prior to photo-activation followed by a time-lapse recording of alternating mCherry and local two-photon excitations. The region for photo-activation was limited to three consecutive z-stacks (1 μm spacing) centered at 4 μm from the apical-most plane. During each round of activation, depending on the size of the region, a z-stack was excited for a total effective illumination time between 152 and 380 ms per pulse, at a laser power of 10 mW. The time between two-photon excitations was between 13 and 24 s.

In order to test the extent to which synthetic furrow could be induced to internalize, to determine the extent of z-spreading of RhoGEF2-CRY2::mCherry recruitment and myosin II accumulation, the same protocol as described above was employed with the following modifications. The ROI was manually adjusted to fit the initial area of activated cells as they moved away from the focal plane and constrict their apical area. As the epithelium folded inwards, the central plane of photo-activation was manually adjusted to fit the first four micrometers from the cells' apical-most plane.

**Induction of pulsatile contractions**. For discontinuous photo-activation, a ROI was activated in three consecutive z-stacks (1 μm spacing) centered at 4 μm from

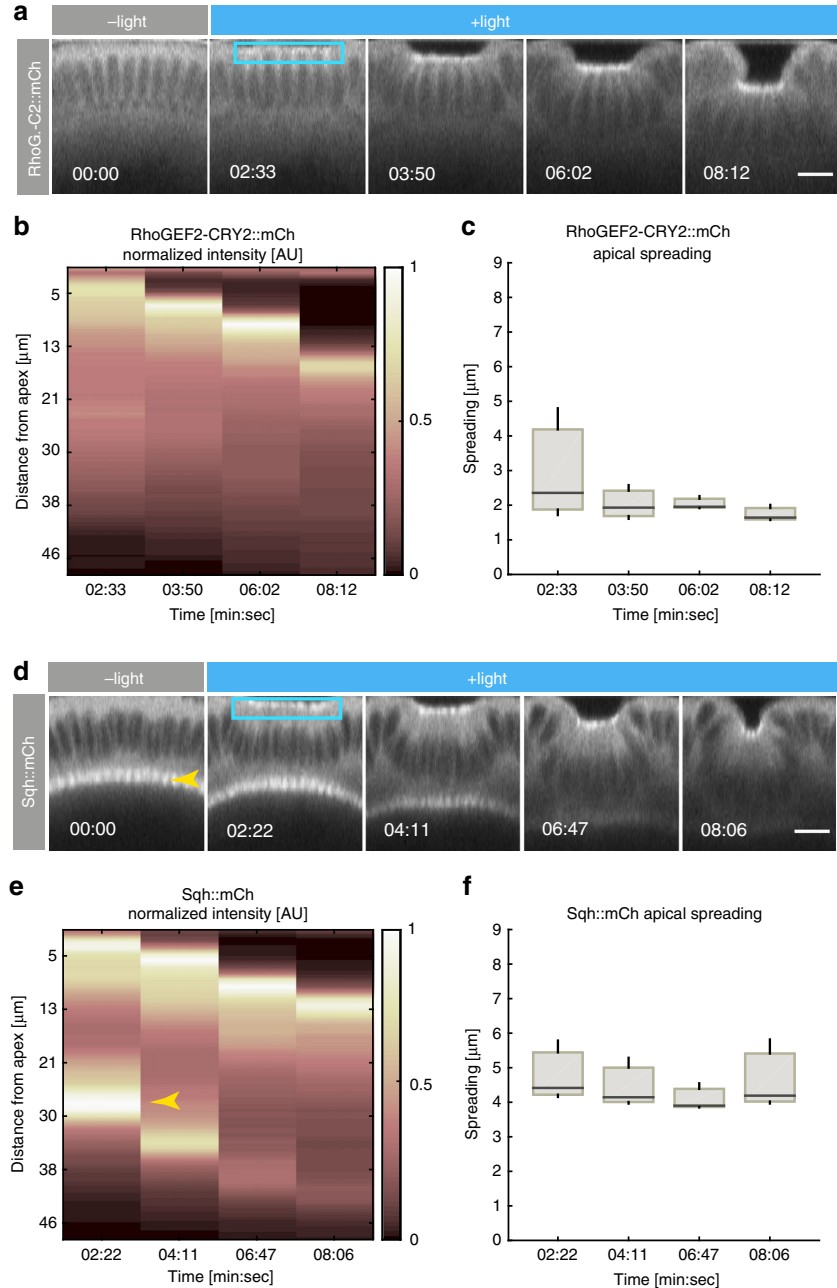

**Fig. 6** Spatial precision of photo-activation (z-spreading). **a**, **d** Confocal still frames from a time-lapse recording of representative embryos ($N = 6$) co-expressing **a** CIBN::pmGFP, RhoGEF2-CRY2::mCherry, or (**d**) CIBN::pmGFP, RhoGEF2-CRY2, and Sqh::mCherry (myosin II). A rectangular region (blue box) spanning $5 \times 15$ cells (XY plane) in three consecutive z-stacks (1 µm spacing) centered at 4 µm from the apical-most plane, was excited for ~228 ms for each acquisition round and alternated with mCherry excitation at intervals of 17 s. Images represents integrated intensity projections of 4 µm of the mCherry signal along the transversal cross-section. Scale bars are 10 µm. **b** Kymograph of RhoGEF2-CRY2::mCherry recruitment along the apico-basal axis of photo-activated cells and (**c**) spreading along the apico-basal axis measured by the Full Width at Half Maximum (FWHM) obtained from Gaussian fitting of the apical intensity signal peak ($N = 3$ embryos). **d** Confocal still frames from a time-lapse recording of a representative embryo co-expressing CIBN::pmGFP, RhoGEF2-CRY2, and Sqh::mCherry. **e** Kymograph of Sqh::mCherry accumulation along the apico-basal axis and (**f**) myosin II accumulation and spreading along the cells' apico-basal axis measured by the FWHM obtained from Gaussian fitting of the apical intensity signal peak ($N = 3$ embryos). Yellow arrowhead in **d** and **e** indicates myosin-II at the basal surface of the cells

the apical-most plane. During each round of activation, the selected region was excited for a total effective illumination time of ~228 ms per pulse, at a laser power of 10 mW. The time between two-photon excitations was 55 s. For continuous photo-activation at a lower laser power, the selected region was excited at a laser power of 5 mW. The time between two-photon excitations was 13 s, the maximum imaging resolution that could be achieved with our system configuration. For testing the effect of photo-activation upon a single illumination pulse, the region of photo-activation was excited for 15 consecutive rounds of two-photon illumination

at a laser power of 15 mW, for a total effective illumination time of ~4 s. Subsequently, in all cases, cell outlines in five consecutive z-stacks (1 µm spacing) were recorded in mCherry during or after photo-activation. For following myosin II dynamics, the dorsal epithelium was subjected to an initial single illumination pulse. Two-photon excitation was restricted to three consecutive z-stacks (1 µm spacing) centered at 4 µm. The region was excited for five consecutive rounds of two-photon illumination at a laser power of 15 mW, for a total effective illumination time of ~2 s. Recordings for both a single plane of two-photon (2 mW) and

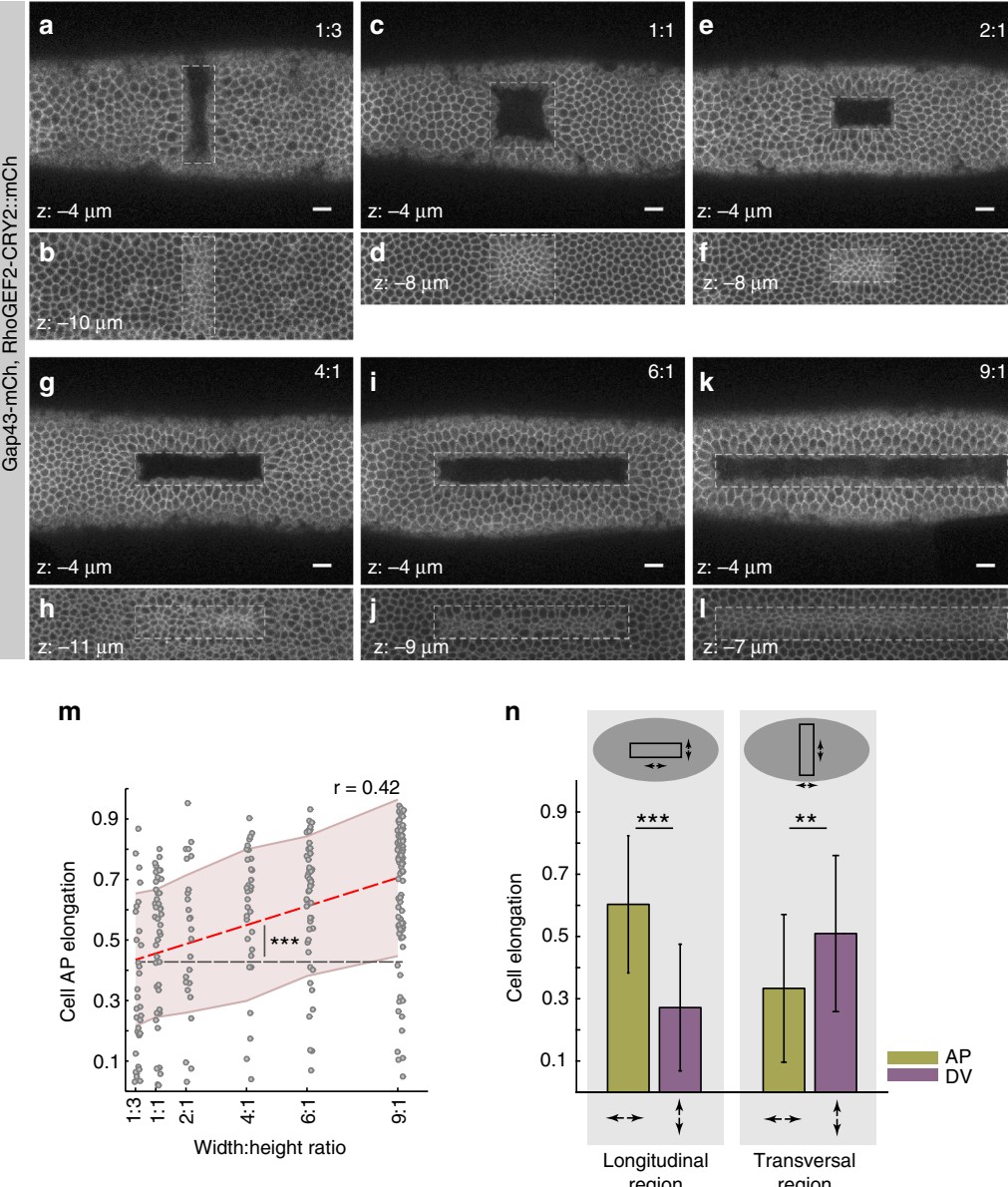

**Fig. 7** The geometry of synthetic contractile domains determines the orientation of cell contraction. **a**–**n** Confocal images of six embryos co-expressing CIBN::pmGFP, RhoGEF2-CRY2::mCherry, and Gap43::mCherry. A single rectangular area of varied width to height ratio (3:1, 1:1, 2:1, 4:1, 6:1, 9:1) was photo-activated on the dorsal epithelium of different embryos. Sustained photo-activation was alternated with mCherry excitation at intervals of ~19 s. After ~6 min, a z-stack was acquired in the mCherry channel to visualize the plasma membrane, note that at the used setting, the RhoGEF2-CRY2::mCherry signal was not visible. Dashed line indicates the area of photo-activation. Scale bars, 10 μm. **a**, **b** Surface views at a depth of 4 and 10 μm for a photo-activation area with a ratio of 1:3. **c**, **d** Surface views at a depth of 4 and 8 μm for a photo-activation area with a ratio of 1:1. **e**, **f** Surface views at a depth of 4 and 8 μm for a photo-activation area with a ratio of 2:1. **g**, **h** Surface views at a depth of 4 and 11 μm for a photo-activation with a ratio of 4:1. **i**, **j** Surface views at a depth of 4 and 9 μm for a photo-activation with a ratio of 6:1. **k**, **l** Surface views at a depth of 4 and 7 μm for a photo-activation with a ratio of 9:1. **m** Quantification of cell elongation along the anterior–posterior embryonic axis (AP elongation, y-axis) in relation to the geometrical pattern of photo-activation measured by the width:height ratio, x-axis. (3:1, $n = 34$; 1:1, $n = 43$; 1:2, $n = 25$; 1:4, $n = 32$; 1:6, $n = 52$; 1:9, $n = 91$). Linear regression analysis reveals a direct relation between cell anisotropy and geometry of activation (red dashed line, slope of 0.032 ± 0.0079, Pearson's correlation coefficient of 0.42, ***p-value of $7.01 \times 10^{-14}$, right-tail t-test against a null hypothesis of a slope equal to zero, black dashed line). Shaded areas indicate root-mean-square deviation (RMSD). **n** Comparison of AP and DV cell elongation between a longitudinal (left) and transversal (right) rectangular pattern of photo-activation. While cell elongation occurs mainly along the AP axis in a longitudinal area of photo-activation, it reorients along the DV axis in the case of a transversal area of photo-activation (*p-value < 0.005, ***p-value < 0.0001, two-tailed paired t-test)

five consecutive z-stacks (1 μm spacing) of mCherry were acquired alternating consecutively in intervals of ~13 s.

**Image and data analysis**. Images were processed and analyzed in MATLAB (MathWorks) using a custom-written pipeline. Images in LSM format were opened and metadata extracted with MATLAB Bioformats toolbox (www.openmicroscopy.

org). For image segmentation, the software Ilastik was used (ilastik.org). The resulting pixel classifier was preserved for images of individual experiments. The generated probability maps were smoothened (size = ~10 px, σ = 2) and thresholded (thr. = 0.5). The binary image was subsequently skeletonized and subjected to watershed transformation. This process resulted in a mask for cell boundaries of 1 px in width. The membrane mask was inverted to obtain masks for individual cells. Cell features such as area and eccentricity, major and minor axis, and

orientation were computed using the function regionprops in MATLAB. Myosin II accumulation pulses were quantified from integrated intensity projections of 5–6 µm for individual cells using the mask from individual cells. We defined cell elongation relative to either AP or DV axis. In brief, AP and DV elongation correspond to the x and y components of the unit vector along the major axis of the cell weighted by cell eccentricity.

Tracking of individual cells was performed by nearest-neighbor assignment, in which the region-overlap between adjacent frames determined the cell identity. For all analysis, we filtered out objects with extreme feature measurements (>2.5 std), as well as object tracks with gaps greater than 2 time points. Myosin intensity for individual cell-masks was measured from integrated intensity projections within 5 µm of the apical-most plane (five consecutive z-stacks, 1 µm spacing). For all measurements of apical area and myosin II accumulation, time-course data was subjected to smoothing by an averaging window of 3 time points. Both instantaneous constriction and myosin accumulation rates were calculated by subtracting consecutive time points and dividing them by their corresponding time interval. For individual pulse identification and cross-correlation analysis, we used the MATLAB functions *findpeaks* and *xcorr2* using the time-course measurements of the constriction and the myosin accumulation rates. For calculating the spreading in the z-axis, we used the full width at half maximum (FWHM) of a Gaussian fitted to a line intensity-plot along the cell's apico-basal axis parameterized by the peak of maximum intensity at the cell's apex.

**Statistical analysis**. Statistical analyses were performed in MATLAB (Math-Works). For determining the relationship between geometry and cell elongation, we first identified the activated cells at the beginning of the protocol and followed them until the formation of a furrow. At that point, we measured their elongation as described in Image and Data Analysis section. We sought for a significant linear regression between the width to height ratio and cell AP elongation using the model:

$$Y = b_0 + b_1 X$$

To test whether the slope of the regression line differs significantly from zero, we used the t statistic defined by:

$$t = \frac{b_1}{SE}$$

where

$$SE(b_1) = \frac{\sqrt{\frac{\Sigma(y_i - \hat{y}_i)^2}{(n-2)}}}{\sqrt{\Sigma(x_i - x)^2}}$$

Finally, we used the tcdf MATLAB function to extract the p-value from the upper-tail probability of t, corresponding to test for a positive slope against a null hypothesis of a slope equal to zero. For testing differences in the orientation of anisotropy upon change in ROI orientation, and for testing the differences in the period durations for the different illumination protocols, we used two-tailed paired Student's t-tests.

**TUNEL and antibody double labeling**. For detection of apoptosis, embryos expressing CIBN::pmGFP, RhoGEF2-CRY2::mCH, and the plasma membrane Gap43::mCh were dechorionated for 2 min in sodium hypochlorite solution. After activation, individual embryos were fixed in 4% paraformaldehyde (Electron Microscopy Sciences) and heptane (Sigma) for 20 min in the dark. The embryos were then devitellinized and kept in methanol at −20 °C. As a positive control for apoptosis, embryos that had passed stage 10 of embryogenesis and had therefore started apoptosis, were dechorionated and fixed as described above. Embryos were blocked in 10% bovine serum albumin (BSA) in PBS, 0.1% Triton-X-100 (Sigma) for 1 h. Fixed embryos were incubated with an anti-GFP antibody (abcam 6556, 1:1000) in PBS containing 5% BSA and 0.1% Triton-X-100 for 2 h at room temperature. Embryos were washed four times in PBS, 0.1% Triton-X-100, and incubated overnight at 4 °C in the dark with Alexa 647 anti-rabbit secondary antibodies (1:500), diluted into the appropriate TUNEL reaction mixture (In Situ Cell Death Detection Kit, Fluorescein, Roche). After four washes in PBS, 0.1% Triton-X-100 and four washes in PBS. For image acquisition, stained embryos were mounted in MatTek dishes covered with PBS and orientated to display the previously activated area.

**Data availability**. The authors declare that all data supporting the findings of this study are available within the article and its supplementary information files or from the corresponding author upon reasonable request.

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

## Acknowledgments

We thank all members of the De Renzis laboratory for helpful discussion and Giorgia Guglielmi for helping during the initial stage of this project. We thank M. Coppey for sharing critical information on the RhoGEF2-CRY2 construct used in this study. We thank D. Arendt, E. Furlong, J. Rink, and A. Runge for critical reading of the manuscript. We thank the advanced light microscopy core facility (EMBL, Heidelberg) for their advice and assistance, and A. Politi for providing the Pipeline Constructor macro. We thank B. Klaus from the EMBL Centre for Statistical Data Analysis for discussions related to the data analysis. We thank the Bloomington Drosophila Stock Center for providing fly stocks and the Drosophila Genomics Resource Center for providing cDNAs.

## Author contributions

The experiments were conceived and designed by E.I. and S.D.R. Cloning, TUNEL assays, and transgenic flies by T.Q. All remaining experiments were performed by E.I. All the data were analyzed by E.I. and S.D.R., who also wrote the manuscript together.

## Additional information

**Competing interests:** The authors declare no competing interests.

