## [Peer Review File · Nature Communications]

Reviewers' Comments:

Reviewer #1:

Remarks to the Author:

In this article, Izquierdo and colleagues use optogenetic methods to investigate the relationship between signaling through the small GTPase Rho, cellular apical constriction, and tissue invagination. The authors use an optogenetic technique that they previously developed to localize RhoGEF2, an upstream activator of Rho, to the plasma membrane. Using this approach, they show that they can induce cells to invaginate in different geometric patterns, in regions of *Drosophila* embryos in which cells do not normally invaginate. Invagination of a patch of tissue in the optogenetic experiments is associated with the apical localization and contraction of the molecular motor myosin II in the cells in which RhoGEF2 was localized to the membrane. A single pulse of RhoGEF2 activation induced coordinated cycles of apical expansion and contraction, but no tissue invagination. In contrast, sustained RhoGEF2 activation for 10 min was sufficient to induce tissue invagination. Using their optogenetic tools, the authors also investigate whether the shape of the invaginating tissue affects the isotropy of the constriction at the individual cell level. They find that, as cells constrict, they elongate parallel to the long axis of the invaginating tissue.

The study is generally well-conducted and pretty convincing. I just have three small questions on controls and data interpretation that I think should be addressed before publishing the work:

MAJOR

1. The authors argue that a single pulse of Rho activation can induce cycles of apical constriction and relaxation but not tissue invagination. They speculate that a single pulse may not be sufficient to establish radial polarity of Rho signaling components. The authors should test this, and also quantify how long did pulsatile contractions last after a single pulse of optogenetic Rho activation.
2. Can the authors show that the cells they are illuminating to induce their apical constriction are not apoptotic?
3. No statistical analysis is shown for any of the quantitative analysis. This would be particularly useful in Figure 4h.

MINOR

1. Page 5: "blue box in Fig. 1a" should be "blue box in Fig. 2a".
2. Panel e in Extended data Fig. 2 should be in lowercase in the figure.

Reviewer #2:

Remarks to the Author:

Epithelial folding mediated by apical constriction provides a common mechanism to convert 2-dimensional epithelial sheets into 3-dimensional, complex tissue structures during embryogenesis. A long-standing question is whether cell sheet folding also requires pre-patterned tissue properties (e.g. mechanical properties) in addition to the contractile forces generated at the apical surface. In this manuscript, Izquierdo et al use state-of-the-art optogenetic approaches to demonstrate that plasma membrane recruitment of the GEF domain of RhoGEF2, an upstream activator of myosin II, is sufficient to induce apical constriction and furrow invagination in early embryonic epithelium that is not normally prescribed to form a furrow. Interestingly, whereas furrow invagination requires sustained recruitment of RhoGFF2, a single pulse of recruitment is sufficient to trigger cycles of apical contraction and relaxation, suggesting that pulsed contractions can arise independent of a temporally patterned input of RhoGEF activity. Finally, the authors demonstrate that the anisotropy of cell constriction is primarily determined by the geometry of the constricting domain, whereas the curvature of the embryo does not have an obvious impact. The authors conclude that

apical constriction induced by the RhoGEF2-Rho pathway is sufficient to trigger epithelial folding independent of the prescribed cell fates or tissue curvature.

The findings provide new insights into the minimal requirements for apical constriction mediated tissue folding and could have important implications for the development of tissue engineering. The optogenetic approach developed in this work would provide a very useful tool for studying the mechanics of apical constriction. I believe the work will be of broad interest to the field of developmental biology and regenerative medicine.

There are some issues about the methods and observations that I hope the authors can clarify:

1. It remains unclear to what extent the recruitment of RhoGEF2 is restricted at the apical domain. The authors use a multiphoton laser to excite RhoGEF2-CRY2 within a confined focal volume of the cytoplasm near the apical cortex, but the activated RhoGEF2-CRY2 should be free to diffuse and may bind to pm-CIB all over the plasma membrane (probably depending on the binding affinity and diffusion rate). The authors should show an image-stack demonstrating the spatial distribution of RhoGEF2 along the apical-basal axis after stimulation. If RhoGEF2 is also recruited to the basolateral membrane, would myosin contractility also be induced at the basolateral membrane? Will the result be different if the tissue is excited with a single-photon laser?
2. The authors suggest that induction of apical constriction is sufficient to induce epithelial folding independent of tissue differentiation, but only present data on the first embryonic tissue in early gastrula. To demonstrate that this conclusion is indeed generalizable, the authors should carry out similar test in some other epithelia, such as in the mid-/late embryonic epithelia.
3. The cyclic contraction and relaxation of cells after a single pulse of RhoGEF2 recruitment is very interesting. The interval between pulses (180 s) is ~ twice as long as that observed in the constricting cells during ventral furrow formation (~90 s, Martin et al., 2009). The authors should discuss the possible reasons for this difference. Does the period of contraction pulses depend on laser intensity or length of induction?
4. According to Movie 4, the induced cells appear to contract rather synchronously after a single pulse of RhoGEF2 recruitment. To what extent the cell behavior can be interpreted by the behavior of apical myosin? Does myosin also undergo cycles of accumulation and dispersion in a synchronized manner across the activated region?

Some minor issues:

It would be helpful to include a diagram demonstrating the domain organization of RhoGEF2, as well as the domains of RhoGEF2 that are included in the fusion construct.

Fig 2b' and c: a figure showing the change of average cell eccentricity should be included.

Page 7, line 2, "Fig g-i" should be "Fig 3. g-i"

Page 24: Last line, the reference Schuh et al 2007 is missing from the reference list

Page 25: Line 2 "Adam Martin???" should be confirmed.

The laser power used for single initial photo-activation should be described in "Methods"

Page 30: "Extended data Figure 2"—there is no extended data figure 1

A recent study (Wagner and Glotzer, 2016) using a similar strategy to activate RhoA and induce

cleavage furrow ingression should be cited.

Reviewer #3:

Remarks to the Author:

De Renzis and colleagues have used optical genetics to assess the sufficiency of apical plasma membrane Rho-GEF activity for inducing various aspects of tissue invagination. The results show that the induced Rho-GEF activity can produce ectopic furrows resembling the natural ventral furrow in a number of ways (including the apical recruitment of myosin, the constriction of single cells, the invagination of these cells, and the stretching of neighbouring tissue regions towards the invagination). These findings complement previous studies of the ventral furrow by showing which aspects of its morphogenesis are specifically due to apical Rho-GEF activity. Effects of the timing of induction and the spatial pattern of induction are also investigated, revealing points of interest to the specialist. Overall, this study is very exciting and is based on a striking dataset. It employs cutting edge molecular manipulation to probe a central model of tissue morphogenesis. The findings and approach should be of general interest.

I have three criticisms/questions.

1. On page 6 the following is stated, "Considering the reversion kinetics of the CRY2/CIB1 system ($T_{1/2} \sim 5$ min.) 29,30, these results suggest that pulsatile contractions can arise independently of Rho signaling pulsation". I do not agree with this conclusion because the results do not exclude a role for a Rho-GAP in producing the pulsatile contractions. This issue could be addressed with revision to the text.

2. For the pulsing behavior induced by a single induction of Rho-GEF activity, it seems that the whole tissue region contracts and expands repeatedly. The authors compare this result with the pulsatile contractions of individual cells of the ventral furrow or amnioserosa. Since these individual cells pulsate asynchronously in the developing tissue, I question whether the induced whole tissue pulsations are comparable (e.g. could the whole tissue pulsations arise from pulling between the induced tissue and surrounding tissue?). Further analyses of the data could help address this issue. Also, a quantitative comparison with cell pulsing, or lack thereof, with continual GEF induction is warranted.

3. The authors show that continuous Rho-GEF activation can induce formation of a U-shaped invagination similar to the ventral furrow. When does the induced invagination become distinct from the normal invagination? This distinction would highlight the stage when factors other than RhoGEF2 begin to play key roles during ventral furrow development. I suggest adding further data to address this point.

Reviewer #4:

Remarks to the Author:

Izquierdo et al. present an interesting manuscript on the use of optogenetics to control a morphogenetic process in developing *Drosophila*. They link RhoGEF2 to a cryptochrome protein that once light stimulated will localize to its binding partner at the plasma membrane. Their selective illumination of apical membranes demonstrates that Rho pathway activation is sufficient for constriction and the invagination process in a scenario that omits other invagination processes.

My comments are:

- The title is perhaps an overstatement. It appears that only one process (invagination) is put under control of optogenetics here.
- Likewise, it is not evident if indeed this work is a demonstration of "concepts of synthetic

biology" (p. 9, conclusions section), as it is merely one protein that is regulated by an external signal... Synthetic biology often refers to networks of interacting proteins and cells that are engineered to function in new ways.

- The manuscript may benefit for a thorough discussion of existing models for functional links between constriction and invagination and how this work fits into a larger context, e.g. also in light of possible forms of invagination that may occur in the absence of apical constriction. This would allow to judge how much new biological information is obtained.
- One central idea of this work appears to be the study of a tissue that is free from "endogenous invagination processes". Equally interesting would be to understand the interplay of these processes with apical constriction.
- Upon pulsed activation of the optogenetic system an oscillatory behavior is observed but this section of the manuscript ends abruptly without further investigation of the process (e.g. of the duration of illumination needed for "stabilization").

Reviewer#1 has found this study to be well-conducted. We thank this reviewer for his/her comments. Our detailed response is listed below in blue, after each of the points that were raised.

Reviewer #1 (Remarks to the Author):

In this article, Izquierdo and colleagues use optogenetic methods to investigate the relationship between signaling through the small GTPase Rho, cellular apical constriction, and tissue invagination. The authors use an optogenetic technique that they previously developed to localize RhoGEF2, an upstream activator of Rho, to the plasma membrane. Using this approach, they show that they can induce cells to invaginate in different geometric patterns, in regions of *Drosophila* embryos in which cells do not normally invaginate. Invagination of a patch of tissue in the optogenetic experiments is associated with the apical localization and contraction of the molecular motor myosin II in the cells in which RhoGEF2 was localized to the membrane. A single pulse of RhoGEF2 activation induced coordinated cycles of apical expansion and contraction, but no tissue invagination. In contrast, sustained RhoGEF2 activation for 10 min was sufficient to induce tissue invagination. Using their optogenetic tools, the authors also investigate whether the shape of the invaginating tissue affects the isotropy of the constriction at the individual cell level. They find that, as cells constrict, they elongate parallel to the long axis of the invaginating tissue.

The study is generally well-conducted and pretty convincing. I just have three small questions on controls and data interpretation that I think should be addressed before publishing the work:

MAJOR

1. The authors argue that a single pulse of Rho activation can induce cycles of apical constriction and relaxation but not tissue invagination. They speculate that a single pulse may not be sufficient to establish radial polarity of Rho signaling components. The authors should test this, and also quantify how long did pulsatile contractions last after a single pulse of optogenetic Rho activation.

We have added two new figures (Fig. 3 and 4) describing in more details the induction of pulsatile contractions upon three different optogenetic activation protocols (discontinuous, continuous at lower laser power, and single pulse at a higher laser power). In Fig. 4, we have analysed myosin II dynamics during pulsations. These new experiments reveal that myosin II undergoes cycles of medio-apical accumulation and dissolution which occur in phase with apical constriction and relaxation. We did not observe myosin accumulating at three-vertices junction as seen in mutants that fail to establish radial polarity of Rho signalling components during ventral furrow invagination (Mason FM et al NCB 2013). We could not analyse directly Diaphanous dynamics, as we could obtain only very few embryos over-expressing Diaphanous-GFP, RhoGEF2CRY2 and CIBN at the same time. Therefore, we did not comment directly on failure to establish radial polarity. Nonetheless the myosin II dynamics and localization presented in Fig.4 suggest that

optogenetic-induced pulsations arise as a consequence of transient myosin II medio-apical accumulation similarly to dorsal closure contractions (Solon J. et al Cell 2009). We have described these results in the text (p.9-10) and in the discussion (p15-16). In general, pulsatile contractions could be followed for ~12 min. before endogenous gastrulation started and cells moved away from the photo-activated area.

2. Can the authors show that the cells they are illuminating to induce their apical constriction are not apoptotic?

To address this point, we have performed TUNEL stainings in embryos which were locally photo-activated to induce an invagination. These new data are presented in Supplementary Fig. 1 and show that invaginating cells are not apoptotic.

3. No statistical analysis is shown for any of the quantitative analysis. This would be particularly useful in Figure 4h.

Statistical analysis for all the data have been included in the Figures and added to the Figure legends.

MINOR

1. Page 5: "blue box in Fig. 1a" should be "blue box in Fig. 2a".

Corrected

2. Panel e in Extended data Fig. 2 should be in lowercase in the figure.

Corrected

Reviewer#2 has also found this study to be well-conducted and of broad interest to the file of developmental biology and tissue engineering. We thank this reviewer for his/her comments. Our detailed response is listed below in blue, after each of the points that were raised.

Reviewer #2 (Remarks to the Author):

Epithelial folding mediated by apical constriction provides a common mechanism to convert 2-dimensional epithelial sheets into 3-dimensional, complex tissue structures during embryogenesis. A long-standing question is whether cell sheet folding also requires pre-patterned tissue properties (e.g. mechanical properties) in addition to the contractile forces generated at the apical surface. In this manuscript, Izquierdo et al use state-of-the-art optogenetic approaches to demonstrate that plasma membrane recruitment of the GEF domain of RhoGEF2, an upstream activator of myosin II, is sufficient to induce apical constriction and furrow invagination in early embryonic epithelium that is not normally prescribed to form a furrow. Interestingly, whereas furrow invagination requires sustained recruitment of RhoGFF2, a single pulse of recruitment is sufficient to trigger cycles of apical contraction and relaxation, suggesting that pulsed contractions can arise independent of a temporally patterned input of RhoGEF activity. Finally, the authors demonstrate that the anisotropy of cell constriction is primarily determined by the geometry of the

constricting domain, whereas the curvature of the embryo does not have an obvious impact. The authors conclude that apical constriction induced by the RhoGEF2-Rho pathway is sufficient to trigger epithelial folding independent of the prescribed cell fates or tissue curvature.

The findings provide new insights into the minimal requirements for apical constriction mediated tissue folding and could have important implications for the development of tissue engineering. The optogenetic approach developed in this work would provide a very useful tool for studying the mechanics of apical constriction. I believe the work will be of broad interest to the field of developmental biology and regenerative medicine.

There are some issues about the methods and observations that I hope the authors can clarify:

1. It remains unclear to what extent the recruitment of RhoGEF2 is restricted at the apical domain. The authors use a multiphoton laser to excite RhoGEF2-CRY2 within a confined focal volume of the cytoplasm near the apical cortex, but the activated RhoGEF2-CRY2 should be free to diffuse and may bind to pm-CIB all over the plasma membrane (probably depending on the binding affinity and diffusion rate). The authors should show an image-stack demonstrating the spatial distribution of RhoGEF2 along the apical-basal axis after stimulation. If RhoGEF2 is also recruited to the basolateral membrane, would myosin contractility also be induced at the basolateral membrane? Will the result be different if the tissue is excited with a single-photon laser?

In order to address this point, we have quantified the localization of both RhoGEF2-CRY2 and myosin II over-time along the apico-basal axis of activated cells. These data are presented in Fig.6 and show that the z-spreading for both RhoGEF and myosin II is limited to 3-5 microns. As this reviewer correctly noted, the extent of z-spreading is a combination of binding affinity and diffusion rate which under the conditions established in this study allow the selective activation of RhoGEF at the apical surface. Localized contractility can be induced also by the selective recruitment of RhoGEF2 to the basolateral surface. However, these data have not been included in this manuscript as they go beyond the scope of this work and they will be the focus of a future analysis.

The reason why we needed to employ two-photon activation is two-fold. First, when using the CRY2/CIB1 system, single-photon activation does not allow precise x-y (single cell resolution) photo-activation, see panels a-b in the figure below and for more details see Guglielmi G. et al Dev Cell 2015. Second, single photon illumination does not allow the selective activation of the apical surface, as illustrated in panel c. We did not include these data in the manuscript, as we felt they were too technical and would have not added sufficient additional biological insights.

2. The authors suggest that induction of apical constriction is sufficient to induce epithelial folding independent of tissue differentiation, but only present data on the first embryonic tissue in early gastrula. To demonstrate that this conclusion is indeed generalizable, the authors should carry out similar test in some other epithelia, such as in the mid-/late embryonic epithelia.

In Supplementary Fig. 2 we show induction of an ectopic invagination during later stages of gastrulation when the embryo has acquired already a more complex morphology.

3. The cyclic contraction and relaxation of cells after a single pulse of RhoGEF2 recruitment is very interesting. The interval between pulses (180 s) is ~ twice as long as that observed in the constricting cells during ventral furrow formation (~90 s, Martin et al., 2009). The authors should discuss the possible reasons for this difference. Does the period of contraction pulses depend on laser intensity or length of induction?

We have added an entire new figure (Fig.3) describing pulsatile behaviour upon three different photo-activation protocols (discontinuous, continuous at lower laser power, and single pulse at a higher laser power). In all cases, the period is longer than the one of ventral furrow cells (~150 s for the discontinuous and continuous at lower laser power). In the discussion we speculate that this difference might be due to a limit imposed by the reversion kinetics of the CRY2/CIB1 system in the dark.

4. According to Movie 4, the induced cells appear to contract rather synchronously after a single pulse of RhoGEF2 recruitment. To what extent the cell behavior can be

interpreted by the behavior of apical myosin? Does myosin also undergo cycles of accumulation and dispersion in a synchronized manner across the activated region?

To address this question, we have analysed myosin II dynamics, see new Fig. 4. Consistently with the prediction made by this reviewer, myosin II undergoes cycles of medio-apical accumulation and dissolution which occur in phase with apical constriction and relaxation.

Some minor issues:

It would be helpful to include a diagram demonstrating the domain organization of RhoGEF2, as well as the domains of RhoGEF2 that are included in the fusion construct.

This information has been added to Fig.1 (panel a)

Fig 2b' and c: a figure showing the change of average cell eccentricity should be included.

A new panel showing average cell eccentricity over-time has been added to Fig. 2 (panel d)

Page 7, line 2, "Fig g-i" should be "Fig 3. g-i"

This has been corrected and updated to the new figure numbering

Page 24: Last line, the reference Schuh et al 2007 is missing from the reference list

This reference has been removed and the correct Drosophila stock number has been reported in the methods.

Page 25: Line 2 "Adam Martin???" should be confirmed.

Updated

The laser power used for single initial photo-activation should be described in "Methods"

Updated

Page 30: "Extended data Figure 2"—there is no extended data figure 1

All extended data are now reported as Supplementary Figures.

A recent study (Wagner and Glotzer, 2016) using a similar strategy to activate RhoA and induce cleavage furrow ingression should be cited.

We have added the missing citation

Reviewer#3 We thank this reviewer for his/her positive comments on our study. Our detailed response is listed below in blue, after each of the points that were raised.

Reviewer #3 (Remarks to the Author):

De Renzis and colleagues have used optical genetics to assess the sufficiency of apical plasma membrane Rho-GEF activity for inducing various aspects of tissue invagination. The results show that the induced Rho-GEF activity can produce ectopic furrows resembling the natural ventral furrow in a number of ways (including the apical recruitment of myosin, the constriction of single cells, the invagination of these cells, and the stretching of neighbouring tissue regions towards the invagination). These findings complement previous studies of the ventral furrow by showing which aspects of its morphogenesis are specifically due to apical Rho-GEF activity. Effects of the timing of induction and the spatial pattern of induction are also investigated, revealing points of interest to the specialist. Overall, this study is very exciting and is based on a striking dataset. It employs cutting edge molecular manipulation to probe a central model of tissue morphogenesis. The findings and approach should be of general interest.

I have three criticisms/questions.

1. On page 6 the following is stated, "Considering the reversion kinetics of the CRY2/CIB1 system ($T_{1/2} \sim 5$ min.) 29,30, these results suggest that pulsatile contractions can arise independently of Rho signaling pulsation". I do not agree with this conclusion because the results do not exclude a role for a Rho-GAP in producing the pulsatile contractions. This issue could be addressed with revision to the text.

We agree with this point and therefore we have removed this statement from the results and we have added the following sentence to the discussion: "Stimulation of Rho signalling above a certain threshold could override, for example, the activity of a RhoGAP, which is required to control the normal spatio-temporal dynamics of Rho GDP/GTP cycling. In agreement with this proposal, pulsatile constrictions during ventral furrow invagination require the activity of a specific RhoGAP⁴⁸."

2. For the pulsing behavior induced by a single induction of Rho-GEF activity, it seems that the whole tissue region contracts and expands repeatedly. The authors compare this result with the pulsatile contractions of individual cells of the ventral furrow or amnioserosa. Since these individual cells pulsate asynchronously in the developing tissue, I question whether the induced whole tissue pulsations are comparable (e.g. could the whole tissue pulsations arise from pulling between the induced tissue and surrounding tissue?). Further analyses of the data could help address this issue. Also, a quantitative comparison with cell pulsing, or lack thereof, with continual GEF induction is warranted.

In Fig. 4, we have analysed myosin II dynamics during pulsations. These new experiments reveal that myosin II undergoes cycles of medio-apical accumulation and dissolution which occur in phase with apical constriction and relaxation. These

new data are compatible with the hypothesis that pulsatile behaviour is induced by myosin cycling. We have made it clear in the text that these pulsations are synchronous and therefore different from ventral furrow or dorsal closure pulsations. Regarding continuous GEF induction, In Fig. 2e,f we have quantified the constriction rate of individual cells and we did not identify any cycle of pulsatile activity suggesting that continuous GEF stimulation induces continuous apical contraction.

3. The authors show that continuous Rho-GEF activation can induce formation of a U-shaped invagination similar to the ventral furrow. When does the induced invagination become distinct from the normal invagination? This distinction would highlight the stage when factors other than RhoGEF2 begin to play key roles during ventral furrow development. I suggest adding further data to address this point.

We agree that it would interesting to explore in more experimental details the difference between optogenetic-guided and endogenous invagination processes. However, given the focus of this present study and the additional experiments included in this revised version, we did not address this point with additional experiments. We have added a paragraph to the discussion (p.14) where we discuss differences with endogenous invaginations: "However, apical constriction is not sufficient to drive closure of an invagination into a tube-like structure, as seen for example during ventral furrow formation. Additional pushing forces exerted by lateral non-invaginating cells and/or loss of myosin II from the basal surface and basal relaxation might be required to complete this step, as suggested by a recent study based on a 2D finite element model³⁰."

Reviewer#4 We thank this reviewer for his/her positive comments on our study. Our detailed response is listed below in blue, after each of the points that were raised.

Reviewer #4 (Remarks to the Author):

Izquierdo et al. present an interesting manuscript on the use of optogenetics to control a morphogenetic process in developing Drosophila. They link RhoGEF2 to a cryptochrome protein that once light stimulated will localize to its binding partner at the plasma membrane. Their selective illumination of apical membranes demonstrates that Rho pathway activation is sufficient for constriction and the invagination process in a scenario that omits other invagination processes.

My comments are:

- The title is perhaps an overstatement. It appears that only one process (invagination) is put under control of optogenetics here.

We agree that this study is focused on only one specific morphogenetic process. However, invagination is a modular process that contributes to the development of a variety of different organisms in different morphogenetic contexts and therefore in our opinion it can be considered as a general example of morphogenesis. Importantly, by reconstructing invagination this study reveals key features common to many morphogenetic processes. For example, we show that by triggering changes at the cell level we can induce changes at the tissue level, and that some of

these changes are emergent features of collective cell interactions (i.e. link between contractility and geometry). On the basis of these considerations, we think that our study does uncover minimal features of tissue rearrangements that can be referred to as morphogenesis and hence the choice of the proposed title.

- Likewise, it is not evident if indeed this work is a demonstration of "concepts of synthetic biology" (p. 9, conclusions section), as it is merely one protein that is regulated by an external signal... Synthetic biology often refers to networks of interacting proteins and cells that are engineered to function in new ways.

We agree that the concepts of synthetic biology borrowed in this study might be obscured by the simplicity of the optogenetic system that we have employed and that this could raise concerns of whether it qualifies as a full-fledged synthetic system. We also agree that in general our optogenetic system lacks the characteristic network of interacting proteins providing designed feedback control loops. However, we do think that our study borrows concepts from synthetic biology at least in two aspects. First, a tunable input-output function is established between the amount of light and the extent of contraction of the tissue (i.e pulsatile, geometrical constraints, etc). These kinds of relationships are common themes in synthetic biology. Second, a common feature of synthetic approaches is to juxtapose natural biological regulation to the precise orthogonal control over signaling pathways and cell behavior (p. 3 ref. 12-17). In a sense, it would be fair to say that cells are engineered to constrict in response to new ways of activation. In any case, to avoid ambiguity we have added a paragraph at the end of the introduction where we explain the meaning of "synthetic" in the context of our study, referring in particular to the nascent field of synthetic morphogenesis.

- The manuscript may benefit for a thorough discussion of existing models for functional links between constriction and invagination and how this work fits into a larger context, e.g. also in light of possible forms of invagination that may occur in the absence of apical constriction. This would allow to judge how much new biological information is obtained.

Page 4 of the introduction is now entirely dedicated to discussion of other existing models of invaginations, which we agree helps to put our work into a more general context.

- One central idea of this work appears to be the study of a tissue that is free from "endogenous invagination processes". Equally interesting would be to understand the interplay of these processes with apical constriction.

We agree that it would be interesting to explore in more detail the interplay between apical constriction and endogenous invagination processes. However, given the scope of this study, we did not address this point in this manuscript.

- Upon pulsed activation of the optogenetic system an oscillatory behavior is observed but this section of the manuscript ends abruptly without further

investigation of the process (e.g. of the duration of illumination needed for "stabilization").

To address this point, we have included new data presented in Fig. 3 and Fig. 4. In Fig. 3, we have examined pulsatile behaviour in response to three different photo-activation protocols (discontinuous, continuous at lower laser power, and single pulse at a higher laser power). In all these cases pulsations were not followed by a stabilization phase of the constricted state. Continuous apical contractions and invagination could only be induced by increasing the laser power with the sustained photo-activation protocol presented in Fig. 2. In Fig. 4, we have analysed myosin II dynamics during pulsations. These new experiments reveal that myosin II undergoes cycles of medio-apical accumulation and dissolution which occur in phase with apical constriction and relaxation.

Reviewers' Comments:

Reviewer #1:

Remarks to the Author:

Well-done review, the original manuscript was already really nice. The authors have addressed my concerns and I support publication.

Reviewer #2:

Remarks to the Author:

The authors have addressed all my previous questions. The new data included in the revised manuscript are elegant and further demonstrate the usefulness of the optogenetic tool developed in this study. I now fully support publishing this nice piece of work in Nature Communication.

Some minor corrections:

Line 142: Supplementary Fig. 2b,d,f should be Supplementary Fig. 1b,d,f.

Line 239: Supplementary Fig. 3 should be Supplementary Fig. 2

Fig. 3, the number of cells seems to be switched between panel a and b, according to the figure legend.

Line 882: "for the same three embryos as in (c)" should be "... as in (b)"

Line 884: An "(e)" should be added after "and corresponding constriction and myosin II accumulation rates".

Line 942-943: "(b) Kymograph of Sqh::mCherry accumulation along the apico basal axis and (c) Myosin II accumulation and spreading along...": (b) and (c) should be (e) and (f), respectively.

Figure 6d and 6e: the meaning of the yellow arrowheads should be explained in the figure legend.

Figure 6b,c,e,f: the label of the x-axis, "Time [min]", should be "Time [min:sec]".

Reviewer #3:

Remarks to the Author:

My past comments have been addressed effectively. Overall, the revisions have improved what was already an interesting paper.

Reviewer #1 and #3 have no further comments. Reviewer#2 has requested a couple of minor corrections which we have addressed.

Line 142: Supplementary Fig. 2b,d,f should be Supplementary Fig. 1b,d,f.
Changed

Line 239: Supplementary Fig. 3 should be Supplementary Fig. 2
Changed

Fig. 3, the number of cells seems to be switched between panel a and b, according to the figure legend.
Changed

Line 882: “for the same three embryos as in (c)” should be “... as in (b)”
Changed

Line 884: An “(e)” should be added after “and corresponding constriction and myosin II accumulation rates”.
Changed

Line 942-943: “(b) Kymograph of Sqh::mCherry accumulation along the apico basal axis and (c) Myosin II accumulation and spreading along...”: (b) and (c) should be (e) and (f), respectively.
Changed

Figure 6d and 6e: the meaning of the yellow arrowheads should be explained in the figure legend.
Done

Figure 6b,c,e,f: the label of the x-axis, “Time [min]”, should be “Time [min:sec]”.
Changed